# Why Is the Australian Health Sector So Far behind in Practising Climate-Related Disclosures?

**DOI:** 10.3390/ijerph191912822

**Published:** 2022-10-06

**Authors:** Tom Deweerdt

**Affiliations:** School of Earth and Environmental Sciences, University of Queensland, Brisbane 4067, Australia; t.deweerdt@uqconnect.edu.au

**Keywords:** disclosures, health sector, Australia, CDP, TCFD, ASX100

## Abstract

The health sector in Australia and the ASX100 is lagging far behind in the implementation of carbon management and climate risk analysis. This case study highlights the low quantity and quality of the sector compared to its market weight. The analysis of CDP disclosures for Australian healthcare companies shows this delay and a general lack of interest in the Task Force on Climate-Related Financial Disclosures’ (TCFD) recommendations. Yet, the physical and transitory risks for these companies do exist. The reasons for this inaction represent a knowledge gap in the literature, but several hypotheses are formulated, such as the lack of pressure from public authorities. At the level of the ten largest healthcare companies in the world, this failure to act is not systemic, so the scope of analysis must be broadened to see a pattern emerging.

## 1. Introduction

The global temperature has risen by about 0.85 °C over the last 130 years. In the last 25 years, the rate of increase has accelerated to more than 0.18 °C of warming per decade. Sea levels are rising, glaciers are melting, and the distribution of rainfall is changing [1]. In addition to these factual events, there are global climate changes, which are more complex to measure, and which manifest themselves—among other things—in extreme weather events (drought, floods, heat waves) that are increasing in intensity and frequency [1]. Apart from a few rare effects of climate change that could be considered positive, such as a decrease in winter mortality in temperate zones, these appear to be deleterious to health [2]. One example is the negative impacts of climate change on the yields of most crops. In their sixth assessment report, the IPCC (Intergovernmental Panel on Climate Change) noted numerous periods of very rapid increases in the price of food and cereals as a result of extreme weather events [2].

This has, and will have, a very strong impact on the poorest populations, for whom buying food has become very difficult. Reports published by the British journal *The Lancet* identified climate change as the greatest global threat to public health in the 21st century [3]. On Thursday 29 November 2018, *The Lancet* published the second edition of its ‘Lancet Countdown’ report dedicated to the health aspects of climate change. A collaboration of 27 academic institutions, the UN and intergovernmental agencies from all continents, the report reveals the unacceptable risk to the current and future health of people around the world as a result of climate change [3]. The effects of climate change on human health are direct and indirect, with all populations at risk [4]. This IPCC carbon footprint target for 2050 would stabilise global warming at 1.5 °C, and would only be achievable by reducing global CO2e emissions from the current 40 Gigatonnes per year to zero [5]. Several signatory states to the Paris Agreement have pledged to meet this target by 2050, including Australia. To do this, local and international corporations, which are major emitters of CO2e, must do their share of the work [6]. Thus, in this objective of reducing emissions, one of the essential elements of the process is to undertake carbon accounting. This allows companies to monitor their CO2e emissions in the three scopes, and to choose emission management techniques accordingly [7]. Unfortunately, some companies are not experts in the field of carbon accounting, and the practice is too often prone to errors that can jeopardise carbon management strategies aimed at Net Zero 2050 targets [8]. In this context, ‘carbon disclosure’ refers to the practice of encouraging or requiring organisations or individuals to report greenhouse gas emissions that may have an impact on the environment [9]. Much of the participation in carbon reporting is voluntary [10]. Another benefit cited for participating in carbon disclosure is an enhanced public image regarding contribution to environmental protection [11].

Carbon disclosure is often seen as the first step in reducing emissions [12]. This step includes setting an organisation’s reduction targets, calculating emissions levels, reporting, and auditing [13]. These disclosures also allow access to the help of certain non-profit organisations such as the Carbon Disclosure Project (CDP). CDP verifies the data of companies that have chosen to disclose, checks the proficiency of the accounting, audits management strategies, and advises companies. Companies can also access strategic carbon and environmental data from other market stakeholders, such as climate-risks disclosures [14].

The health sector in Australia is not identified by the TCFD as being highly exposed to climate change risks, and is therefore referred to as a “non-key TCFD sector” [15]. Within the Australian Stock Exchange (ASX), companies are increasingly making climate-related disclosures (Figure 1), but it seems that the health sector is far behind, as only two ASX100-listed companies have provided disclosures to the CDP since 2010 [14]. The health sector accounts for 7% of national emissions in Australia [16]. Given the low level of disclosure by the sector, there is a major need for improvement [17].

The literature shows the importance of carbon risk disclosures in companies. Companies’ Corporate Social Responsibility (CSR) and the willingness to disclose is linked to specific determinants. For Australian companies, the literature highlights company size, industry sector, profitability, and corporate governance mechanisms as major determinants [18]. Companies are also committed to AGENDA 2030 and the UN Sustainable Development Goals. In Australia, global action is needed to achieve these goals [19]. The literature therefore highlights the major role of the health sector in these considerations [20]. Additionally, Environmental, Social and Governance (ESG) reporting has been linked to a firm’s social license to operate because it affects a firm’s reputation [21]. Firms lacking ESG practices are perceived as having poor performance [22]. Reputation is essential in the healthcare industry because it influences trust and translates into intent to purchase health-related products [23]. The healthcare industry has a delay in the implementation of their environmental practices [24]. ESG reporting in disclosures from the healthcare industry is recent [25]. Biotech and pharmaceutical firms use important volumes of toxic chemicals in production [26] and emit more greenhouse gases than the automotive industry [24]. Climate change is likely to force millions of people to migrate, which could lead to an increase in mental health problems and increased pressure on health infrastructure. There is also a risk of increased food shortages and the spread of certain diseases [27]. The main risk for the health sector is that the burden on the sector will increase further as global temperatures rise, increasing its own greenhouse gas emissions [28]. Climate change therefore poses significant risks to the healthcare industry within its supply chain [29]. 

For these reasons, the focus of this research is on Australian health sector carbon disclosure practices. The aim of this research is to highlight the flaws in the Australian health sector’s development of climate change risk mitigation strategies, particularly in the disclosure process. Nevertheless, as only two major health companies are providing disclosures in Australia, an intersectoral comparison is essential. This comparison will help to highlight fundamental differences and delays between the biggest firms and sectors in Australia, by looking at the best practices and performance of certain sectors against the health sector. This will show the great gap of the Australian health sector in disclosures. The materials used for this research are, in order of importance, the scientific literature on best practice in climate disclosure; the specificity of the Australian health sector, particularly in relation to carbon management and the supply chain; and the risks to companies associated with climate change, particularly in relation to health. The second main data source for this research is the CDP’s reports on the climate disclosures of Australian companies in the ASX100. Closely related to this data, the research used government and private sector reports (e.g., from the Australian government and EY). Finally, the research is based on the TCFD’s recommendations on climate disclosures.

## 2. Methods

### 2.1. Methodology and Definition of the Case

The methodology used for this research is the intrinsic case study methodology. In the intrinsic case study, theories are mobilised to analyse and understand the case studied; this assumes that the case itself carries a ‘theory’ or analysis. In other words, the situation analysed must fall within a class of management problems that can be identified and from which different concepts and theories are mobilised [30]. The research is based on the analysis of climate disclosures of health sector companies in the ASX100, Australia. The six companies in question are: ANSELL—Richmond, Australia; Coachlear—Sydney, Australia; CSL—Hongkong, China; Ramsay Healthcare—Sydney, Australia; Resmed Inc.—San Diego, CA, USA, and Sonic Healthcare—Sydney, Australia. Nevertheless, and given the low number of disclosures made by these companies, the analysis of climate disclosures was conducted on the CDP results from 2018 to 2021 for ANSELL and CSL. The other five firms studied did not provide any disclosures. Beyond this data, the case study focuses on comparing the Australian health sector of the ASX100 with other sectors in the country, namely: telecommunications and IT, consumer staples, real estate, financials, energy, materials, industrials, utilities, and consumer discretionary.

### 2.2. Selection of the Case

The decision to select the implementation of climate disclosures in the Australian health sector was made due to the uniqueness of the subject. There are questions about the delay in disclosure by health companies. The sector is indeed non-key in terms of its impact on the environment, according to the TCFD, but the disruptions arising from climate change are particularly impactful for the health sector. This case selection aims to understand the mechanisms related to carbon accounting and disclosure in the health sector. The comparison between the health sector and competing sectors on the stock market is intended to highlight the environmental underperformance of the health sector compared to others.

### 2.3. Analysis, Interpretation, and Reporting

The analysis of the performance of the Australian health sector concerning their carbon and climate-related disclosures has been carried out through rigorous analysis of available disclosures, other government and private data and reports, and by highlighting patterns identified in the scientific literature. Comparison of the quantity and quality of disclosures with other stock market sectors is essential. The sectoral comparison takes into account the environmental performance of firms in the sector, based on the quality and quantity of disclosures made against their weight on the Australian stock market. If the environmental scores and the weight on the stock market are estimated as a percentage, the ratio between the two is estimated as follows: ratio = environmental performance percentage/weight in the ASX100.

## 3. Results

On the ASX100, the Australian stock market, only six companies are in the health sector [31]. Their total market capitalisation is AUD 1,401,564,248,027 and their market share amounts to 10.33% (Stock Metric, 2022). The sector is strongly led by CSL, which is the country’s third largest listed company. CSL is an Australian medical company specialising in vaccines, anti-poisons, and blood products. Its stock has doubled since 2018 (Stock Metric, 2022). 

Healthcare contributes about 7% of Australia’s CO2e emissions, including the country’s public hospitals and health system [16]. This is relatively small compared to the share of the Australian stock market taken by companies in the sector. Nevertheless, all companies must reduce their emissions to meet the targets set by the Australian government. The first step in the strategy to reduce emissions and mitigate the impact of climate change is reporting and disclosure [13].

The TCFD is developing best practice in the area of disclosures, and considers the health sector in Australia to be non-key [15]. The recommendations are developed into four categories: governance, strategy, risk management, and metrics and targets [32]. The health sector in Australia is known to be the worst performer in the environmental reporting and disclosure system [15,28,33]. Of the six largest companies in the sector, only two provided disclosures to the CDP: CSL—the largest—and ANSELL—the smallest (Table 1). This is very few when you consider that ASX100 companies have globally doubled their number of climate-related disclosures since 2017 (Figure 1). Furthermore, the CDP scores for these two companies are globally low, based on their rating indexes, ranging from A to F (Table 1). Except for ANSELL from 2018 to 2019, there has been no improvement in the ratings of the two companies. Even worse, CSL has regressed in 2021 from a C to a D (Table 1). 

This shows a disinvestment by CSL in their climate change mitigation and emissions abatement strategy. Indeed, companies that report to CDP usually use the organisation’s reports and recommendations to improve their strategy [34]. The healthcare industry has poor climate change reporting practices, only one company committed to setting climate change targets, pledges are rare, and no targets of the Paris Agreement are mentioned through their reporting. 

The environmental performance related to disclosures of the ASX100 sectors, based on an analysis of the quality and quantity of company disclosures, shows that the health sector is far behind its competitors (Figure 2).

The score is only 3.55%, while some smaller sectors, such as telecommunications, have more than 10 times this figure. 

By comparing the percentage of environmental performance with the weight of the sectors within the ASX100, a ratio can be derived. This ratio further clarifies the differences between the sectors, and the results can be analysed as follows: a ratio between 0 and 1 shows a poor environmental performance compared to the weight of the sector in the market. A ratio of 1 would mean a performance equal to the weight of the sector. A ratio greater than 1 indicates that the sector has a higher environmental performance than its weight in the ASX100.

Thus, it is evidenced that the health sector in Australia is far behind, with a ratio of 0.344. The only three sectors below the threshold of 1 are health, finance, and materials (Table 2). 

ANSELL and CSL covered less than 6 of the 11 TCFD recommendations, and the disclosures were focused on the targets and metrics area, with risk management disclosures being neglected. Overall, the quality of CDP climate change disclosures from 2018 to 2020 was poor, with the strategic aspects showing the lowest quality [35,36].

ANSELL’s 2021 disclosures show that the company has no emissions targets [35].

Scope 3 includes all other indirect emissions that occur in a company’s value chain [37]. The methodology for calculating Scope 3 indirect emissions for ANSELL is inadequate, resulting in an inability for the company to account for Scope 3 emissions [35]. Indirect Scope 3 emissions are, however, very important—especially for pharmaceutical companies, as waste-related emissions are part of this scope [37]. The health industry has a large amount of waste [38]. The literature shows that Scope 3 is very often neglected, even though a large amount of emissions result from it [39]. In carbon accounting best practice, the correct calculation of Scope 3, its inclusion, and its abatement are essential [40,41]. ANSELL therefore does not follow best practices in this area.

The company claims to be very communicative, but has not made a pledge. Additionally, ANSELL indicates that it does not engage with its value chain on climate-related issues [35]. Finally, the company indicates that it is strongly opposed to national and local regulations. The company implemented two abatement actions, and saves only 538 metric tons of CO2e per year [35]. At the level of the risks foreseen by the company, the risk assessment methodology used is not sufficiently advanced to perceive the potential risks. The methods used by ANSELL and CSL are far from the recommendations and frameworks of the TCFD or the Global Programme on Risk Assessment and Management for Adaptation to Climate Change [42,43]. Even if these risks are less significant than those faced by other sectors, the impact of climate change on biomedical activities is real [15,44].

CSL is the only company that has made a pledge to move towards a low-carbon strategy. Nevertheless, this transition plan announced by the company is not a scheduled resolution item. This means that the decision to follow this plan is not formally a company decision. In terms of the influence of climate change risks and opportunities on the company’s strategy, CSL has not assessed any influence on products and services, investment, or R&D and operations [36]. The only category influenced is the supply chain, with two opportunities found, but no risks. The company implemented five abatement actions and saves only 608 metric tons of CO2e per year, or 0.3% of their total disclosed emissions [36]. The company has not calculated its Scope 3 emissions, despite the recommendations of the CDP for several years. The reductions in activities are considered to be decreasing [36].

According to the TCFD’s risks categories: “Transitioning to a lower-carbon economy may entail extensive policy, legal, technology, and market changes to address mitigation and adaptation requirements related to climate change. Depending on the nature, speed, and focus of these changes, transition risks may pose varying levels of financial and reputational risk to organizations” [42]. 

Physical risks resulting from climate change can be event-driven (acute) or longer-term (chronic) in climate patterns. Physical risks may have financial implications for firms and organizations in the health sector, such as direct assets threats or indirect impacts for their supply chain [45]. Extreme temperature changes might affect the Australian health sector’s premises, operations, and transport needs [46].

## 4. Discussion

The literature argues that the health industry, particularly the pharmaceutical industry, is at risk from climate change [47,48,49]. The sector’s exposure to environmental hazards is, according to several studies, very high [48,50,51]. The pharmaceutical sector is already bearing the brunt of this risk, with high financial losses, such as in Puerto Rico in 2017, where Hurricane Maria washed away nearly USD 200 million worth of pharmaceutical products from Pfizer warehouses [52]. In addition, heat and water stresses are a considerable risk for the biotechnology industry [53]. Indeed, India produces nearly 50% of the world’s vaccine demand, and is at high risk from water and heat [53]. The literature also shows that these risks are not sufficiently considered by companies [48,54,55]. Their management and mitigation are therefore severely limited. One of the reasons outlined in the literature is the low quantity and quality of climate risk disclosures within the sector [56]. Another reason is the limitation of financial metrics in the disclosure frameworks related to Nature, such as water or heat [57]. Given the complexity and diversity of the health sector’s activities and its supply chain, climate risks need to be highlighted to a far greater extent [15]. The literature therefore calls for companies to improve the disclosure process, but also for international institutions to develop a new framework for Nature-related physical risks [58]. This is happening, with the creation of the Taskforce for Nature-related Financial Risks Disclosures (TNFD).

Another potential reflection of the literature regarding the assessment of the quality and quantity of disclosures within the CDP is the presence of limitations within the TCFD and CDP frameworks [59]. Indeed, the literature argues that climate disclosures are in fact a means for companies to deal with national or international regulations and enhance their communication [59,60,61]. Thus, disclosures are self-regulating by sector or by country, like a financial market [62]. Moreover, the CDP’s climate disclosures are not concrete enough to allow for evaluation of the environmental or climate performance of an entire sector [63]. The literature also argues that the greater the number of disclosures made at an early stage of a company’s environmental strategy, the more effective and useful they are [62,63]. As the pharmaceutical and health sector lags, these disclosures were not made at an early stage. 

Apart from the risks of climate change for this sector, some companies are thinking about the potential opportunities of climate change [64]. Indeed, climate change will lead healthcare companies to increase their activity through new opportunities, as was the case with the COVID-19 epidemic [65]. It is therefore not necessarily in the interest of healthcare companies to act immediately to reduce emissions and slow down the process [66]. As a reminder, the US company Pfizer increased its stock market index by 65.4% between July 2020 and July 2022 [67].

### 4.1. Hypotheses on the Reasons Why the Health Sector Discloses Very Little on Climate Change

One of the first assumptions about why the health sector is lagging in climate disclosure is philosophical. The health industry believes it is doing the greater good by developing and providing health products for the general public [28]. In terms of social disclosures, the Australian health sector is also lagging, and ranked 6th out of the 10 stock market sectors. This shows a real reluctance to provide information [33].

A second assumption is that social and governmental pressure is limited. Few incentives and obligations exist to push industry leaders to disclose [28].

The health sector is bound to secrecy of information to develop their products (like drugs or vaccines) and then patent them extensively [68]. This could lead to an intrinsic reluctance to the practice of disclosure in general. Public health is also subject to an omertà where disclosing patients’ personal data is simply forbidden. Many people working in public health then decide to join the private sector and are used to this silence [69].

### 4.2. Is There a Similar Pattern in Other Global Markets?

Of the ten largest healthcare companies in the world, nine submit their disclosures to the CDP each year. Roche Holding-Genusschein is the only company that has declined to respond. In addition, the majority of the other nine companies received a grade of B and then A. This runs counter to the pattern identified in the Australian market. CSL, the largest Australian company in the sector, is only 23rd in the world [70]. In Bangladesh, different studies stated that the pharmaceutical industry’s climate change disclosure performance was still at an initial stage [55]. Broadening the scope of research and analysis is therefore essential to ensure that this pattern exists globally. 

## 5. Conclusions

The case study of Australian stock market health companies showed that climate risk mitigation strategy and GHG emissions abatement were not the priority of the sector. It ranked last among the country’s nine other stock market sectors. Yet, the share of these companies in the ASX100 is very high. Only two companies chose to respond to CDP’s requests for disclosure. The quality of the data disclosed is generally low, and the CDP ratings follow this trend. The disclosures of the two companies are not sufficient to draw conclusions regarding the sector as a whole, but it is worth noting that the strategic aspect, one of the pillars of the TCFD, is left out. The same applies to Scope 3 emissions accounting. In parallel to these findings, Australian healthcare companies do not pledge to reduce emissions or achieve Net Zero; the targets set are only related to Scope 1 intensity, and are almost non-existent. When comparing the sector’s environmental performance against its weight in the ASX100, the results are even weaker. The health sector does not appear to see many climate-related risks; however, according to the TCFD definitions of these risks, the diversity of health activities and the complex supply chain are indeed at risk, even if these risks are lower than for other sectors and classified as non-key by the TCFD. The health sector, on the other hand, has many opportunities to take advantage of the impact of climate change on human health and to further develop their activity. The reasons for the low quantity and quality of climate disclosures from the sector are mainly based on assumptions; the research on the topic is limited, and should be enhanced. Indeed, the financial risks are real and already negatively impact the whole sector, but the focus might be moved onto Nature-related risks, which lack metrics and framework. To motivate the sector to disclose about Nature physical risks, research on the field should be improved. Government pressure is limited and the pharmaceutical industry believes it is doing the public good and is built around a taste for secrecy and the extensive use of patents. In global markets, this pattern does not seem to hold true for the ten largest healthcare companies, so the scope of analysis must be broadened.

## Figures and Tables

**Figure 1 ijerph-19-12822-f001:**
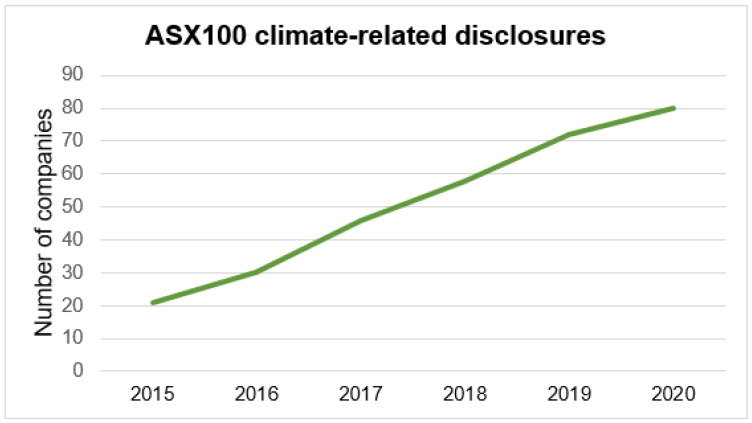
ASX100 Australian climate-related disclosures evolution (Symons 2021).

**Figure 2 ijerph-19-12822-f002:**
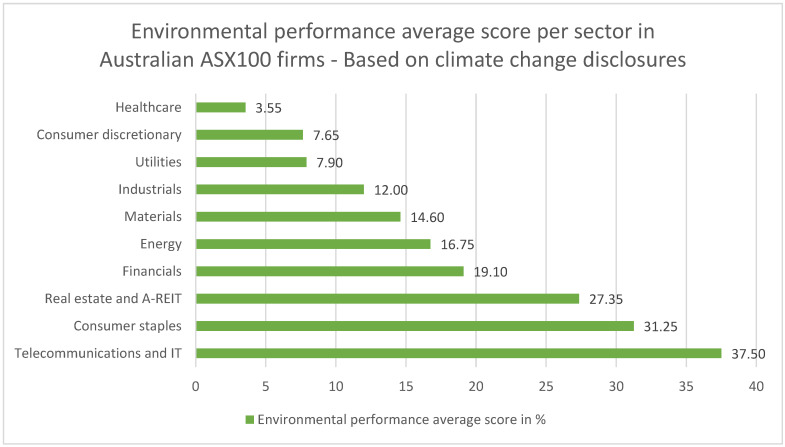
Environmental performance average score per sector in Australian ASX100 firms based on climate change disclosures (source: ACCA, 2011).

**Table 1 ijerph-19-12822-t001:** ASX100 healthcare companies’ disclosures and scores.

Company	ANSELL	Coachlear	CSL	Ramsay Health Care	Resmed Inc.	Sonic Healthcare
Disclosures						
**Climate Change 2021**	Submitted B	Submitted Not Available	Submitted D	No Response	No Response	No Response
**Climate Change 2020**	Submitted B	No Response	Submitted C	No Response	No Response	No Response
**Climate Change 2019**	Submitted B	No Response	Submitted C	No Response	No Response	No Response
**Climate Change 2018**	Submitted C	No Response	Submitted D	No Response	Declined to Participate	No Response

**Table 2 ijerph-19-12822-t002:** Weight and environmental performance ratio and ranking in ASX100 sectors.

SECTORS	Weight in the S&P/ASX100 (In %)	Environmental Performance Based on Climate Change Disclosures (In %)	Ratio between Weight and Environmental Performance per Sector	Ratio Ranking
FINANCIALS	30.54	19.10	0.625	8
MATERIALS	17.77	14.60	0.821	7
HEALTHCARE	10.33	3.55	0.344	9
INDUSTRIALS	8.45	12.00	1.420	6
REAL ESTATE	6.90	27.35	3.964	3
TELECOMMUNICATIONS AND IT	6.27	37.50	5.981	1
CONSUMER DISCRETIONARY	6.13	7.65	1.248	
ENERGY	6.08	16.75	2.755	5
CONSUMER STAPLES	5.50	31.25	5.682	2
UTILITIES	2.07	7.90	3.816	4

## Data Availability

Not applicable.

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
