# Peer review of "Why Is the Australian Health Sector So Far behind in Practising Climate-Related Disclosures?"

_ijerph, 2022, doi:10.3390/ijerph191912822_

Round 1

Reviewer 1 Report

Dear Authors

Your article is interesting, I am grateful for the opportunity to read it. I think that idea and the subject are very interesting.

Reading the text, I found some elements those I think would improve your article.

An abstract is information about an article, its content, goals, assumptions, method used, results and conclusions. A good abstract structure is needed by the reader, but it is also important for you as authors - readers often use the abstract review method to search for content that interests them. Your abstract is not well-structured, and this may be a barrier to popularizing your article.

Introduction - The introduction should provide not only a research background but also an indication of a precise research/review objective, the research questions and methods used. Adding a brief indication of the logic of presenting the research material in the last paragraph of the introduction is highly recommended.

You do not have a "literature review" chapter, it is difficult to read an article when you do not know exactly what the main purpose of the article is, what are the main assumptions and what literature is the basis for the research. In my opinion it is mandatory to separate introduction from literature review.

Methodology - The choice of the case appears to be entirely voluntary, and no source or study confirms this.
[89-90] The
decision on selecting the implementation of climate disclosures in the Australian Health sector was made due to the uniqueness of the subject.

There is no information about the methods used to prepare the comparison as well as the whole article is missing. This information is insufficient. In addition, there are also no sources.
[108-109] The analysis of the environmental performance of the Australian health sector has been carried out through rigorous analysis of available disclosures, other government and private data and reports, and by highlighting patterns identified in the scientific literature.

Due to lack of explanation and lack of sources. This whole part seems to be completely voluntary and based only on your intuition

Discussion and Conclusions - To increase the relevance of the results, the discussion section should cover the differences and similarities between your findings and that of other researchers. I miss it here, so I think you should create a part where you truly present the comparison between your findings and opinions and findings of other researchers. Your proposition is too poorly sourced.

The concluding part should be a short summary of the goals, methods, and findings of the article. These chapters should be expanded. For me, these parts (Discussion and Conclusions) is too limited, there is too modest reference to your assumptions and your research questions.

Summarizing.

I really like your article and appreciate your work. It is interesting topic, and the conclusions could open the way for further research.

But without a clear and precisely formulated background, a well-built literature database, and wide comparison to different research, this kind of articles will be always less assessed. If you want to write good article you should consider it.

Overall, I am very impressed because your article is very interesting.

Good luck!

Author Response

Dear Reviewer, 

Thank you very much for your recommendations, a lot of revisions have been made to the manuscript. 

Best regards

Tom Deweerdt

Reviewer 2 Report

Please refer to the attached file for comments on the study. Good luck

Author Response

(The authors gave the same response as above.)

Round 2

Reviewer 2 Report

The author addressed all the doubts that emerged from the previous version and filled in the gaps reported. The discussion of the results is still not particularly incisive, the added value of the study and its implications could be better highlighted. However, the current version of the paper is certainly more effective than the previous one, clearer and more interesting to read. It could be accepted in the current version (no methodological errors or conceptual misunderstandings were detected) but I would only recommend further strengthening the discussion to increase the impact of the research

Author Response

Thank you for all your precious recommendations,

Best regards
